# Prevalence of Type 2 Diabetes, Impaired Fasting Glucose, and Diabetes Risk in an Adult and Older North-Eastern Portuguese Population

**DOI:** 10.3390/healthcare11121712

**Published:** 2023-06-11

**Authors:** Pedro M. Magalhães, José E. Teixeira, João P. Bragada, Carlos M. Duarte, José A. Bragada

**Affiliations:** 1Research Centre in Sports Sciences, Health and Human Development (CIDESD), 5001-801 Vila Real, Portugal; jose.eduardo@ipb.pt (J.E.T.); jbragada@ipb.pt (J.A.B.); 2Department of Sport Sciences, Instituto Politécnico de Bragança (IPB), 5300-253 Bragança, Portugal; a36348@alunos.ipb.pt; 3Department of Sport Sciences, Instituto Politécnico de Guarda (IPG), 5300-253 Guarda, Portugal; 4North East Local Health Unit (ULSNE), Health Care Unit of Santa Maria, 5301-852 Bragança, Portugal; joao.bragada@ulsne.min-saude.pt

**Keywords:** diabetes, prediabetes, community sample, regional, epidemiology

## Abstract

The aims of this study were (1) to evaluate the prevalence of type 2 diabetes (T2D) in a middle-aged north-eastern Portuguese population, (2) to analyze the prevalence of impaired fasting glucose (IFG), and (3) to assess the risk of T2D in this community-based sample. An exploratory, retrospective, and cross-sectional study was conducted from a total of 6570 individuals aged 18–102 years, among which 3865 were women (57.4 ± 18.1 years) and 2705 were men (60.0 ± 16.8 years). T2D diagnosis, IFG, and the diabetes risk score (low to very high risk) were assessed. The prevalence of T2D in this adult and an older north-eastern Portuguese population was 17.4%. A higher prevalence of T2D was reported in men (22.2%) than in women (14.0%); however, this was without significant differences (*p* = 0.086). Otherwise, the prevalence of T2D was significantly different among the age groups and increased with age (*p* < 0.001). Regarding IFG, a higher percentage of cases was observed in men (14.1%) than in women (8.4%) (*p* < 0.001). The risk of developing T2D in the next 10 years showed an association with sex and age group (*p* < 0.001) with a small-to-moderate effect (V = 0.1–0.3). Men and the elderly had the highest percentage of cases in the moderate-to-very high-risk bands. The current research confirmed a higher prevalence of T2D, IFG, and diabetes risk than previous Portuguese epidemiological reports. The results also suggest potential prediabetes cases, which should be carefully monitored. The current research adds evidence to the worldwide trend of the increasing prevalence of T2D and intermediate hyperglycemia (i.e., prediabetes).

## 1. Introduction

Obesity is increasing worldwide and is one of the main causes of type 2 diabetes (T2D), which is a chronic metabolic disease with a growing impact on morbidity and mortality [1,2]. T2D is characterized by a multiple etiology with a loss of organic glucose homeostasis and insulin effectiveness, with implications also at the level of carbohydrate, fat, and protein metabolism [3,4]. However, until now, no effective prevention methods have been adopted [5,6]. These metabolic disorders are the consequence of a relative or absolute deficiency in insulin secretion by the pancreatic β-cells, as well as a greater or lesser resistance to this hormone by the target cells [7,8]. Chronic hyperinsulinemia is reported as the process that prevents the complete decompensation of glucose homeostasis; however, this compensatory response of pancreatic β-cells is not without consequences [9]. This phenomenon is described by several authors as the silent degradation of insulin effectiveness, referred to as the prediabetes state or intermediate hyperglycemia. Although, for several years, the increase in fasting or maintained postprandial hyperglycemia may not occur while the pancreatic β-cells function responds [10,11]. Hence, people with prediabetes usually have an increased blood glucose concentration which, when coupled with abnormalities of carbohydrate, fat, and protein metabolism, leads to a variety of macro and microvascular complications [12,13,14]. The prediabetes condition is a precursor of T2D, described as borderline diabetes, and this is reversible in contrast to diabetes [10,15].

Prediabetes and diabetes diagnostic criteria were defined according to health authorities, specifically the American Diabetes Association (ADA) [16], the National Institute for Health and Care Excellence (NICE) [17], the World Health Organization (WHO), and the International Diabetes Federation (IDF) [18,19]. Identifying and controlling prediabetes cases becomes fundamental to decreasing the incidence of T2D and, subsequently, their long-term complications [15]. The long-term complications related to diabetes can be subdivided into two complementary levels: (i) microvascular complications, such as peripheral neuropathy, nephropathy, and retinopathy, (ii) macrovascular complications, including cardiovascular disease reflected by major adverse cardiovascular events (MACE), such as amputation, dialysis, blindness, stroke, coronary sclerosis, myocardial infarction, angina pectoris, and impaired circulation of the lower limb [12].

A high worldwide prevalence of T2D has been reported in developing and developed countries, expecting that 439 million adults will be diagnosed with diabetes by 2030, which represents a 69% increase in cases between 2010 and 2030 [20]. Until now, the IDF report estimated a total of 451 million adults worldwide with a diabetes diagnosis in 2017 and projected an increase to 693 million by 2045. In Europe, this incidence rate represents 16% more if the 2017 data is taken into account, from 58 to 67 million people [21,22]. In Portugal, previous studies also detected a high prevalence of T2D; however, the latest Portuguese epidemiological reports about diabetes were performed in 2010, 2015, and 2018 [23]. The first diabetes prevalence study in Portugal, entitled PREVADIAB study, estimated a prevalence of T2D at 11.7 %. The latest epidemiological data reports an increase of 13.6% in the prevalence of T2D in the Portuguese population, whereas 5.9% of these cases were undiagnosed [23,24,25]. Therefore, the latest epidemiological data from 2018 demonstrated that 28.0% of the Portuguese adult population aged between 20 and 79 years has intermediate hyperglycemia, which represents a total of 2.1 million people [23].

Based on global estimates of the prevalence of diabetes for 2030, this disease shows an exponential increase worldwide, particularly in developed countries such as Portugal [20]. Therefore, an update on the epidemiological situation of diabetes in the Portuguese population is needed, and it is essential to understand the specific epidemiological status of each particular region. Thus, the aims of this study were (1) to evaluate the prevalence of T2D in a middle-aged north-eastern Portuguese population, (2) to analyze the prevalence of impaired fasting glucose (IFG), and (3) to assess the risk of T2D in this community-based sample. It was hypothesized that there would be an increase in the prevalence of T2D and prediabetes in this Portuguese population when compared to previous epidemiological reports.

## 2. Materials and Methods

### 2.1. Research Design and Sample

A community-representative sample was selected for this retrospective, observational, and cross-sectional analysis from patients’ clinical records of two primary healthcare centers in a north-east Portuguese region [26,27]. A total of 6570 individuals aged 18–102 years were analyzed between January 2019 and December 2020, among which 3865 were women (57.4 ± 18.1 years), and 2705 were men (60.0 ± 16.8 years). A total of 18,890 individuals were analyzed, with 12,320 of them being excluded from the data analysis based on the following inclusion criteria: (i) patients with an age of 18–102 years old; (ii) individuals with complete clinical information, without the omission of data regarding complementary diagnostic exams, as well as a personal and family history of cardiovascular and metabolic pathology. The exclusion criteria considered for data screening were (i) patients with age < 18 years; (ii) missing information from clinical criteria: height, weight, body mass index (BMI), and demographic considerations. When calculating the sample size with a margin of error of 0.05 and a 95% confidence interval (CI), the minimum sample size should be 383 individuals (G*Power, version 3.1.5.1 Institut für Experimentelle Psychologie, Düsseldorf, Germany). Table 1 presents the descriptive statistics for sampled population according to sex.

### 2.2. Ethical Procedures

The current research was integrated into an exercise-based intervention project in Bragança County for chronic non-communicable diseases [26]. Ethical procedures were ensured in accordance with the guidelines for scientific research on human beings in the Nuremberg Code and the Declaration of Helsinki [28]. This scientific research was approved by the Ethics Committee of the Northeast Local Health Unit of the Santa Maria Health Unit (CNPD no. 2020/149).

### 2.3. Data Collection

#### 2.3.1. Anthropometric Measures

Anthropometric measures were retrospectively evaluated from the patient’s clinical records, specifically weight (kg), height (cm), and WC (cm) [29]. BMI was calculated by dividing weight (kg) by the square of height (m). European BMI cut-offs were used to define normal (<25.0 kg/m^2^), overweight (25.0 to 29.9 kg/m^2^), and obesity (≥30.0 kg/m^2^) [30].

#### 2.3.2. Age Groups

Subjects were divided into three age groups in agreement with standard recommendations: young adults (18–39 years), middle-aged adults (40–64 years), and older adults (≥ 65 years) [26]. The most representative age group was the older adults, with 2776 (42.3%), followed by middle-aged adults, with 2688 (40.9%) individuals. Additionally, men had the highest number of older people (*n =* 1240, 45.8%); however, middle age was the most representative group in women (*n =* 1594, 41.2%). Young adults were the least prevalent age group in both sexes, with 371 (13.7%) men and 735 (19.0%) women.

#### 2.3.3. Diabetes Diagnosis and Fasting Glucose

T2D diagnosis was based on the WHO and IDF criteria (WHO, 2006). T2D was diagnosed in the presence of one of the following criteria: fasting plasma glucose ≥126 mg/dL (≥7.0 mmol/L), classic symptoms and/or oral glucose tolerance test (OGTT) (2 h plasma glucose) ≥200 mg/dL (≥11.1 mmol/L) [31,32]. Blood samples were collected in the morning following standard laboratory procedures and routine enzyme methods to collect 10 mL of venous blood [33]. Fasting glucose (FG), IFG, and T2D diagnosis were confirmed from patients’ clinical records with follow-up in primary healthcare.

#### 2.3.4. Diabetes Risk

The risk of developing T2D in the next 10 years was calculated using the diabetes risk score developed in the FINDRISK tool by Lindström and Tuomilehto [34]. This questionnaire is accepted by the WHO [35] and, in the particular case of the Portuguese population, by the Directorate-General of Health (DGS) [36]. Diabetes considered the following risk bands: low risk (less than 7 points), slight risk (between 7 and 11 points), moderate risk (between 12 and 14 points), high risk (between 15 and 20 points), and very high risk (more than 20 points) [34,35,36]. A total of 4284 individuals had a T2D risk register, of which 1664 (38.8%) were women, and 2620 (61.2%) were men.

### 2.4. Statistical Analysis

For descriptive statistics, the Kolmogorov–Smirnov and Levene’s tests were used to assess normality and homogeneity. Data were presented as mean ± one standard deviation (SD), percentage (%), and 95% confidence intervals (CI). Categorical variables were expressed using counts (*n*) and proportions (%). Chi-squared test or Fisher’s exact test was applied whenever appropriate [37]. Effect size was calculated by Cramer’s V using the following magnitude bands: small (0.1–03), medium (0.3–0.5), and large (V ≥ 0.5) [38]. The modified Wald method was applied to compute the proportion at 95% CI [39]. The prevalence of T2D and IFG was analyzed using a binary logistic regression (log-binary model), with age and sex adjustments to calculate the odds ratio (OR) and their 95% CI. Adjusted OR were performed to express different risk factors and the probability of getting each clinical condition [39,40]. Statistical significance was set at *p* < 0.05. All other statistical analyses were conducted using IBM SPSS Statistics for Windows, Version 27.0 (Armonk, NY, USA: IBM Corp). Data visualizations were computed using Microsoft Excel (Microsoft Corporation, Redmond, WA, USA).

## 3. Results

### 3.1. Prevalence of T2D

The prevalence of T2D in a middle-aged north-eastern Portuguese population was 17.4%, reporting a higher prevalence for men (22.2%) than for women (14.0%). There were no statistically significant differences between both sexes (*p* = 0.086). Additionally, T2D prevalence was significantly different among age groups, increasing with age (*p* < 0.001) with a medium effect (V = 0.3).

Table 2 presents the prevalence of T2D in the studied sample population. T2D prevalence represented 1.2%, 11.9%, and 29.2% of the cases in young adults (18–39 years), middle-aged adults (40–64 years), and older adults (>64 years), respectively. When analyzing the prevalence of T2D in each age group according to sex, women had 0.7%, 8.3%, and 26.4% diabetic cases in the age groups 18–39 years, 40–64 years, and >64 years, respectively. On the other hand, men had 2.2%, 17.1%, and 32.7% of the diagnosed T2D for the same age groups.

### 3.2. Prevalence of IFG

The prevalence of IFG in this community sample was 10.8%. Regarding IFG, a higher percentage of IFG cases was present in men (14.1%) rather than in women (8.4%) (Table 3). When analyzing the prevalence of IFG in each age group according to sex, women had 1.0%, 6.5%, and 14.0% of the diabetic cases in the age groups 18–39 years, 40–64 years, and >64 years, respectively. Otherwise, men had 2.4%, 6.5%, and 14.0% of the diagnosed IFG for the same age groups. The prevalence of IFG was significantly different among sexes and age groups, increasing with age (*p* < 0.001) with a small-to-medium effect (V = 0.2 and V = 0.3, respectively). An age-related increase was observed for IFG, with a greater number of cases being observed in older adults (15.7%) than in middle-aged and young adults (9.5% and 1.4%, respectively) (Table 3).

Regarding IFG, when the diagnosis of T2D was not confirmed, 8.3% of the individuals without diabetes presented some degree of change in fasting glucose in the blood. Women had abnormal values for fasting glucose at 1.0%, 5.3%, and 10.3% for the 18–39 years, 40–64 years, and >64 years age groups, respectively. Conversely, men presented 1.9%, 13.0%, and 15.0% IFG prevalence for the same age groups.

### 3.3. T2D Risk in Individuals without Diabetes

Figure 1 presents the risk of developing T2D in the next 10 years with an association with sex (*p* < 0.001) having a small effect (V = 0.1). Men showed a higher number of cases of slight risk, with 698 individuals, followed by low risk (*n* = 524) and moderate risk (*n* = 296). A high and very high risk of T2D was observed in 200 and 10 individuals, respectively. The same trend was observed in the most prevalent risk bands in women: low risk (*n* = 653), slight risk (*n* = 1160), moderate risk (*n* = 491), high risk (*n* = 334), and very risk (*n* = 29).

Figure 2 shows the risk of developing T2D in the next 10 years within those age groups (*p* < 0.001) with a moderate effect (V = 0.3). We found that the “low risk” score decreases with age, with a value of only 4.1% in the older adult group. The moderate, high, and very high-risk scores increase sharply from young adults to middle-aged adults.

## 4. Discussion

The aims of this study were (1) to evaluate the prevalence of T2D in a middle-aged north-eastern Portuguese population, (2) to analyze the prevalence of IFG, and (3) to assess the risk of T2D in this community-based sample. The research findings have demonstrated a higher prevalence of T2D, IFG, and diabetes risk than previously reported in Portuguese epidemiological studies, highlighting the global trend toward diabetes in developed countries. The Northeast region of Portugal is located inland. It is a region with low population density and little industrialization when compared to the coast of the country. The total population has declined in recent years (from 2011 to 2021, 759 fewer residents for a total of 34,582), particularly in the 15–64 age group. However, the elderly population (>64 years) increased significantly (from 2011 to 2021, with 1543 more residents for a total of 9748) [41]. In the recent past, the per capita income has also been decreasing in this region compared to the European average (78.6% in 2019, 76.4% in 2020, and 74% in 2021), with the north-eastern region of the country showing a lower per capita purchasing power compared to the national average. This reality favors the prevalence of a large number of families in a situation of economic and social vulnerability, aggravated by the confinements motivated by the COVID-19 pandemic, with negative consequences for promoting people’s health [41].

### 4.1. Prevalence of T2D

The prevalence of T2D observed in this exploratory and preliminary analysis was 17.4%. A higher prevalence of T2D was reported in men (22.2%) than in women (14.0%); however, this was without significant differences (*p* = 0.086). Otherwise, the T2D prevalence was significantly different among age groups with an age-related increase (*p* < 0.001) (Table 2). The PREVADIAB study estimated a prevalence of T2D at 11.7 %, slightly lower than the 17.4% that was found in the present study [42]. On the other hand, they also reported a difference between sexes, with a higher prevalence of T2D in men (14.2%) compared to women (9.5%). The current analysis shows the same trend but with higher proportions for both sexes (22.2% in men; 14.0% in women) when compared to the PREVADIAB study. If we consider the data from the first National Health Examination Survey (INSEF), collected in 2015, the overall prevalence of diabetes was lower than that found in the current research (9.8% vs. 17.4%) and in the PREVADIAB study (11.7% vs. 17.4%) [24,25]. However, the trend towards a greater number of subjects living with T2D among males was congruent in the three studies. These sex differences, which have also been observed in the studies reviewed above, may be due to the fact that men develop diabetes at a lower level of abdominal obesity than women [43]. On the other hand, the differences in the prevalence of T2D observed between these studies could be explained by considering some methodological differences between the studies, the target population, and the sampling error associated with prevalence estimates. However, all these studies show that the prevalence of T2D is high in Portugal, being higher than the estimated prevalence of diabetes in Europe reported by the IDF, which is around 9.1% [44].

The present study also shows an increase in the prevalence of T2D with age; that is, 1.18%, 11.87%, and 29.18% of cases in young adults (18–39 years), middle-aged adults (40–64 years), and older adults (>64 years), respectively (Table 2). When analyzing the prevalence of T2D in each age group according to sex, women had 0.7%, 8.3%, and 26.4% diabetic cases in the age groups 18–39 years, 40–64 years, and more than 64 years, respectively. Otherwise, men had 2.2%, 17.1%, and 32.7% of the diagnosed T2D for the same age groups (Figure 1). When comparing these results to the PREVADIAB and INSEF studies, it is possible to observe slightly different proportions, but with an equal tendency for the prevalence of T2D with increasing age [24,42]. This relation may be partially associated with the influence of the senescence process and reduced cellular functionality associated with aging, namely a decrease in the effectiveness of insulin and blood glucose regulation processes, which are widely described in the literature [45,46]. Genetic but also environmental influences, such as behavioral risk factors, which can be changed, should not be neglected either [47]. The study by Gardete-Correia et al. [42] also showed a T2D prevalence of 2.4% in the population between 20 and 39 years old, 12.6% between 40 to 59 years old, and 26.3% in those between 60 and 79 years old. In the same sense, Barreto et al. [24] also reported a higher prevalence of T2D in older age groups, specifically 19% in the 55–64 years old group and 23.8% in the 65–74 years old group. Indeed, most rates in epidemiology and demography are strongly age-dependent (e.g., incidence, prevalence, and mortality) [18,48,49]; therefore, the findings of this study are congruent with the reported age-related impact.

On the other hand, the higher prevalence of T2D in the age group over 64 years old, associated with the fact that this population increased significantly in this region, despite the total population reduction, may also have contributed to the higher prevalence of T2D observed in the present study compared to those described above.

### 4.2. Prevalence of IFG

The prevalence of IFG in this community sample was 10.8% (Table 3). More particularly, 8.3% of the individuals without diabetes presented some degree of change in fasting glucose in the blood. Of those, a higher percentage of IFG cases were present in men (14.1%) rather than women (8.4%). These relative frequencies of IFG are lower than those presented by the PREVADIAB and INSEF studies [24,42]. Gardete-Correia et al. [42] showed the prevalence of prediabetes in the population was 23.3%. However, this percentage includes subjects with IFG (8.2%), plus those with impaired glucose tolerance (IGT) (12.6%) and with both abnormalities of glucose homeostasis (2.4%). When coupled with a T2D diagnosis, the PREVADIAB study estimated that one-third (34.9%) of the adult Portuguese population was affected by diabetes or intermediate hyperglycemia [42]. Likewise, the INSEF study demonstrated that the prevalence of prediabetes was 16% and verified a higher prevalence among women than men (17.5% vs. 14.4%) [24]. However, it is important to note that these data were estimated based on glycated hemoglobin A1c (HbA1c) values between 5.7% and 6.5%. These results are congruent with those of the current research since women have a higher number of cases with some changes in fasting glucose when compared to men (18.0% vs. 12.8%) [24].

When considering the relative distribution of IFG in individuals without diabetes, men have a higher proportion of cases. Moreover, gender differences were followed by an age-related increase in IFG (Table 3). Women had abnormal values of 1.0%, 6.5%, and 14.0% IFG cases in the 18–39 years, 40–64 years, and >64 years age groups, respectively. Conversely, men presented 2.4%, 13.8%, and 17.9% of the diagnosed IFG in the same previous age groups (Figure 2). These data have the same tendency as previous studies, which reported a greater intermediate hyperglycemia condition with advancing age, with a dangerous prognostic trend towards diabetes [10,11,15]. Thus, potential prediabetes cases were observed in this observational cohort sample, having regard only to the IFG values, and should therefore be carefully monitored for the effective primary prevention of diabetes. However, it is important to bear in mind that there are different stages of prediabetes: (i) IFG, which corresponds to altered FG in a blood sample analysis: (ii) IGT, which corresponds to reduced tolerance in dealing with glucose overload, assessed by an oral glucose tolerance test (OGTT) for 2 h plasma glucose (2hPG) after ingesting 75 g of glucose, (iii) both IFG and IGT abnormalities [42,43]. Physical activity and exercise play a key role in the primary and secondary prevention of diabetes, and therefore healthy lifestyle strategies should be implemented to increase their practice in cases of prediabetes [26].

### 4.3. Diabetes Risk in Individuals without Diabetes

When the FINDRISK score is applied, the risk of developing T2D appears to be higher for men than women, specifically moderate risk (18.4% vs. 17.2%) and high risk (12.2% vs. 10.6%). Previous reports in other specific populations had demonstrated an increased risk of T2D for women by applying the FINDRISK score [35,49]. This diabetes risk score has shown validity for predicting the development of T2D in the next 10 years [34]. While being accepted by the WHO [35] and implemented by the Directorate-General of Health (DGS) [36], there are still no validation studies of this diabetes risk score for the Portuguese population in particular. Additionally, the fact that not all of the clinical records present risk calculation may have biased the results. When considering all the components of metabolic syndrome (MetS), in which the IFG precursor of T2D was included, the risk is 1.5 higher for men in comparison to women [27]. Alongside obesity, the mechanisms of insulin resistance that occur in T2D play a key role in the development of atherosclerotic cardiovascular diseases [26].

### 4.4. Limitations, Practical Application, and Futures Perspectives

The interpretation of the results should take into consideration the age groups in the examined sample cohort (i.e., young adults, middle-aged adults, and older adults). Another very important point to consider when interpreting the results is that this research is based on a regional community-representative sample [27]. Moreover, assessing insulin effectiveness and intermediate hyperglycemia, like IGT, would require other diabetes-related parameters, such as OGTT to 2hPG or HbA1c (not considered by the WHO/IDF criteria) [10,11]. The prediabetes diagnosis is also dependent on the criteria used, considering the different cut-off values between the WHO/IDF, ADA, and NICE definitions [15,17,18,19]. ADA [16] defines the normality for FG levels at less than to 100 mg/dL (<5.6 mmol/L), whereas the WHO [19] and NICE [17] do not report minimum cut-off points for normality. As well, IFG is referred if between 100–126 mg/dL (≥5.6 to <7.0 mmol/L) and 110–126 mg/dL (≥6.1 to <7.0 mmol/L), if the ADA or WHO/IDF and NICE criteria were considered, respectively [10,11]. For diabetes, when using venous plasma concentration, the diagnostic criteria considered are the following: (i) FG ≥126 mg/dL (≥7.0 mmol/L); (ii) classic symptoms and/or occasional blood glucose ≥200 mg/dL (≥11.1 mmol/L); (iii) blood glucose ≥200 mg/dL (≥11.1 mmol/L), on OGTT with 75g glucose at 2 h (note that in the absence of unequivocal hyperglycemia associated with classic symptoms, these criteria should be confirmed at a later stage); (iv) impaired glucose tolerance (IGT) <126 mg/dL (<7.0 mmol/L), and, at 2 h, (OGTT) ≥140 and <200 mg/dL (≥7.8 and <11.1 mmol/L) and fasting glycemia abnormality at IFG ≥110 and <126 mg/dL (≥6.1 and <7.0 mmol/L) [15,19]. The differences in the cohort values may be predisposed to different diabetes prevalences [15,17,18,19]. Therefore, the present results should be applied considering the use of the WHO and IDF criteria. Furthermore, in future studies, the risk of developing T2D should analyze the risk factors inherent in the FINDRISK score point by point [34]. In addition, more epidemiological reports with clinical data from other health centers across the country are needed to provide an update on the epidemiological situation of diabetes in Portugal since the current research only updates data in the studied region.

The results of the present study should be interpreted with the following limitations: (i) this is an observational study of a specific region of Portugal; hence, the results cannot be generalized to the rest of the Portuguese population or to other geographic regions; (ii) the lack of access to socio-demographic variables and objective data on the physical activity levels limits a greater depth in the interpretation of the results; (iii) a cross-sectional research design does not allow for a temporal relationship to be established between disease and their risk factors.

## 5. Conclusions

The prevalence of T2D in this adult and older northeast Portuguese population was 17.4%. In trend, men have a higher prevalence of T2D than women, with an age-related increase for both sexes. Furthermore, men have a higher risk of developing T2D within 10 years when using the FINDRISK score, with a considerable increase from the age of 40. The results also suggest potential cases of prediabetes, which should be carefully monitored. Finally, the current research reported a higher prevalence of T2D than previous Portuguese epidemiological reports.

## Figures and Tables

**Figure 1 healthcare-11-01712-f001:**
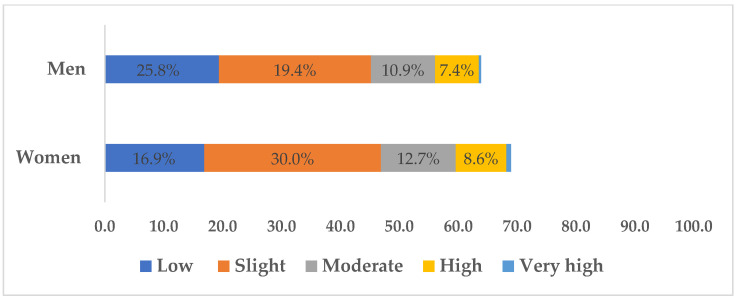
Prevalence of risk of developing T2D in the next 10 years in individuals without diabetes by sex.

**Figure 2 healthcare-11-01712-f002:**
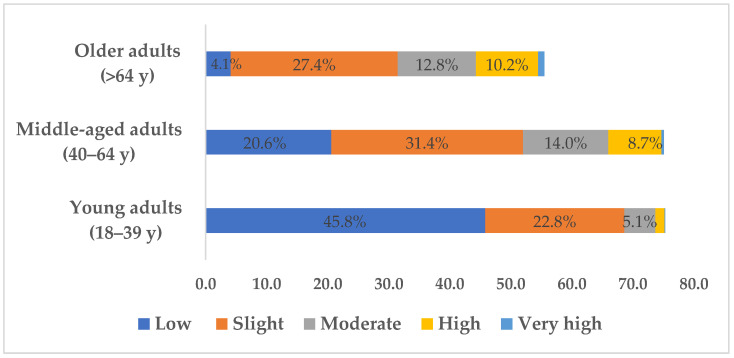
Prevalence of risk of developing T2D in the next 10 years in individuals without diabetes by age group.

**Table 1 healthcare-11-01712-t001:** Descriptive statistics for sampled population according to sex.

Measures	Women (*n* = 3865)	Men (*n* = 2705)	Total (*n* = 6570)
Age (y)	57.4 ± 18.1	60.0 ± 16.8	58.4 ± 17.6
Height (cm)	170.0 ± 7.2	158.7 ± 6.7	163.3 ± 8.9
Weight (kg)	68.7 ± 13.9	80.5 ± 13.5	73.6 ± 14.9
BMI (kg/m^2^)	27.3 ± 5.3	27.8 ± 4.1	27.5 ± 4.9
WC (cm)	92.0 ± 13.3	99.6 ± 11.1	95.2 ± 13.0

Abbreviations: BMI—Body mass index; WC—waist circumference; y—years.

**Table 2 healthcare-11-01712-t002:** Prevalence of T2D in the studied Portuguese population sample according to sex and age group.

Variables	Group Analysis	Diagnosis	*n* (%)	95% CI (Min–Max)
Sex	Women	With T2D	542 (14.0)	(13.0–15.2)
		Without T2D	3323 (86.0)	(84.8–87.1)
		Total	3865 (100.0)	–
	Men	With T2D	600 (22.2)	(20.6–23.8)
		Without T2D	2105 (77.7)	(76.2–79.4)
		Total	2705 (100.0)	–
Age group	Young adults (18–39 y)	With T2D	13 (1.2)	(0.64–2.0)
		Without T2D	1093 (98.8)	(98.2–99.5)
		Total	1106 (100.0)	–
	Middle-aged adults (40–64 y)	With T2D	319 (11.9)	(10.6–13.1)
		Without T2D	2369 (88.1)	(86.9–89.4)
		Total	2688 (100.0)	–
	Older adults (>64 y)	With T2D	810 (29.2)	(27.5–30.9)
		Without T2D	1966 (70.8)	(69.1–72.5)
		Total	2776 (100.0)	–
Population	Overall	With T2D	1142 (17.4)	(16.5–18.3)
		Without T2D	5425 (82.6)	(81.6–83.5)
		Total	6570 (100.0)	–

Chi-square test (X^2^) was used to compare T2D between sexes [X^2^ (1) = 2.9; *p* < 0.086; Cramer’s V = 0.1] and age groups [X^2^ (2) = 345.4; *p* < 0.001; Cramer’s V = 0.3]. Abbreviations: CI—confidence interval; T2D—type 2 diabetes; y—years.

**Table 3 healthcare-11-01712-t003:** Prevalence of IFG in the studied Portuguese population sample according to sex and age group.

Variables	Group Analysis	Diagnosis	*n* (%)	95% CI (Min–Max)
Sex	Women	IFG	326 (8.4)	(7.6–9.4)
		Normal FG	3539 (91.6)	(90.6–92.4)
		Total	3865 (100.0)	–
	Men	IFG	382 (14.1)	(12.8–15.5)
		Normal FG	2323 (85.9)	(84.5–87.2)
		Total	2705 (100.0)	–
Age group	Young adults (18–39 y)	IFG	16 (1.4)	(0.8–2.3)
		Normal FG	1090 (98.6)	(97.7–99.2)
		Total	1106 (100.0)	–
	Middle-aged adults (40–64 y)	IFG	255 (9.5)	(32.9–36.4)
		Normal FG	2433 (90.5)	(89.3–91.6)
		Total	2688 (100.0)	–
	Older adults (>64 y)	IFG	437 (15.7)	(14.4–17.2)
		Normal FG	2339 (84.3)	(82.9–85.6)
		Total	2776 (100.0)	–
Population	Overall	IFG	708 (10.8)	(10.0–11.6)
		Normal FG	5862 (89.2)	(88.5–90.0)
		Total	6570 (100.0)	–

Chi-square test (X^2^) was used to compare IFG and normal FG between sexes [X^2^ (1) = 53.5; *p* < 0.001; Cramer’s V = 0.1] and age-groups [X^2^ (2) = 176.0; *p* < 0.001; Cramer’s V = 0.2]. Abbreviations: CI—confidence interval; IFG—impaired fasting glucose; FG—fasting glucose; y—years.

## Data Availability

Data are available upon request to the contact author.

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
