# Peer review of "Prevalence of Type 2 Diabetes, Impaired Fasting Glucose, and Diabetes Risk in an Adult and Older North-Eastern Portuguese Population"

_healthcare, 2023, doi:10.3390/healthcare11121712_

Round 1

Reviewer 1 Report

x

Suggestions for improvement

Paper is good, but need some language revision.

1. Introduction

Line77

Nevertheless, based on a larger number of previous estimations, a growing burden  of diabetes was established, particularly in developing countries as Portugal [20].

4. Discussion

Line 273

 Research findings have demonstrated a higher  prevalence of T2D, IFG, and diabetes risk than previously reported in Portuguese epidemiological studies, highlighting the global trend toward diabetes in developed countries.

Comment: Portugal is a developing country or developed country?

-        Line 289- 290

This sex differences, which has been observed also in the studies reviewed above, may be due to the fact that men develop diabetes at a lower level of abdominal obesity than women.

Comment: this expectation can be studied by if there was any correlation with waist circumference.

4.4. Limitations, practical application and futures perspectives

Comment: we recommend that this study can be more representive for Portugal if including a sample from different regions.

Paper is good, but need some language revision.

Author Response

Dear Reviewer,

We are also thankful to the reviewers for reviewing the manuscript and suggesting valuable amendments for helping us to enhance its quality. We have revised the manuscript and providing a point to point response to all the comments. Also, we have highlighted all the amends on the manuscript.

Response to Reviewer 1 Comments

Point 1: Paper is good, but need some language revision.

Response 1: Thank you for your comment. We have made best efforts to revise the English language in this new version of the manuscript.

Point 2: 1. Introduction, Line77 Nevertheless, based on a larger number of previous estimations, a growing burden of diabetes was established, particularly in developing countries as Portugal [20]. 4. Discussion, Line 273 Research findings have demonstrated a higher prevalence of T2D, IFG, and diabetes risk than previously reported in Portuguese epidemiological studies, highlighting the global trend toward diabetes in developed countries.

Comment: Portugal is a developing country or developed country?

Response 2: Thank you for pointing out this incongruity. Portugal is a developed country. According to the Human Development Index (HDI), published by the United Nations Development Program for 2021, Portugal ranks 38th out of 190 countries in the world. The HDI takes into account several factors, including life expectancy, education, and income, to assess a country's level of development. We changed the text as follow (lines 77–78):

“Based on global estimates of the prevalence of diabetes for 2030, this disease shows an exponential increase worldwide, particularly in developed countries such as Portugal [20].”

Point 3: Line 289- 290 This sex differences, which has been observed also in the studies reviewed above, may be due to the fact that men develop diabetes at a lower level of abdominal obesity than women.

Comment: this expectation can be studied by if there was any correlation with waist circumference.

Response 3: Thanks for your comment. It is an interesting suggestion, however the aim of this study was to analyze the prevalence of type 2 diabetes (T2D), impaired fasting glucose (IFG) and diabetes risk in a middle-age North-East Portuguese population using a observational and cross-sectional analysis. For this reason, we have applied the following epidemological statistical: (1) categorical variables were expressed using counts (n) and proportions (%); (2) chi-squared test or fisher exact test were applied whenever appropriate (with Cramer’s V for effect size); (3) binary logistic regression (log-binary model), with an age and sex adjustments to calculate odds ratio (OR) and their 95% CI. Adjusted OR were performed to express different risk factors the probability of getting each clinical condition. Otherwise, chi-squared test can be used to test for adherence, homogeneity and independence. That is, chi-square and odds ratio are themselves measures of association and effect size. However, we understand the reviewer's concern that we only use gender and age groups as independent variables, and it is necessary to include other dimensions (socio-demographic, for example) for a more complete characterisation.

Thus, we leave this concern of the reviewer in the research limitations: “…(i) this is an observational study of a specific region of Portugal, hence the results cannot be generalized to the rest of the Portuguese population or to other geographic regions; (ii) the lack of access to socio-demographic variables and objective data on the physical activity of the subjects in the samplelevels, limit a greater depth in the inter-pretation of the results; (iii) cross-sectional research design does not allow a temporal relationship to be established between disease and their risk factors”. Also, we have added the sample power sample power is ensured. “When analysing the sample size, with a margin of error of 0.05 and a 95% Confidence Interval (CI), the minimum sample size should be 383 individuals (G*Power, version 3.1.5.1 Institut für Experimentelle Psychologie, Düsseldorf, Germany). The population living in this region of Portugal comprises 122833 people, based on the 2021 census (PORDATA https://www.pordata.pt/censos/resultados/emdestaque-braganca-446) (reference 41).

Point 4: 4.4. Limitations, practical application and futures perspectives

Comment: we recommend that this study can be more representative for Portugal if including a sample from different regions.

Response 4: We agree with the reviewer opinion. However, the aim of this reserch was to study the prevalence of type 2 diabetes (T2D), impered fasting glucose (IFG) and the risk of T2D in this specific population, to contribute as a baseline for local health policymakers to choose the most appropriate strategies for reversing the trend in the prevalence of T2D. This study complies with the IDF Guidelines, which refer to the need for studies for the planning of care and preventive services. We think that studies that highlight a more local and up-to-date reality can induce greater proactivity on the part of those who make decisions in this matter.

We added the following text with the limitations of the study: “The results of the present study should be interpreted with the following limitations: (i) this is an observational study of a specific region of Portugal, hence the results cannot be generalized to the rest of the Portuguese population or to other geographic regions; (ii) the lack of access to socio-demographic variables and objective data on the physical activity levels, limit a greater depth in the interpretation of the results; (iii) cross-sectional research design does not allow a temporal relationship to be established between disease and their risk factors”. 

For more details please see the revised version manuscript.

Reviewer 2 Report

Dear authors,

Congratulations on your paper.

The topic is of great importance to health professionals, and your research provides a representative sample from Northeast Portugal.

However, I highly recommend that you incorporate additional statistical tests to improve both your confidence in the interpretation and the presentation of results. There are many epidemiological studies available in the literature about the Type 2 Diabetes topic, therefore, the additional analysis would highlight the novel contributions of your paper.

I also suggest addressing the limitations of the analysis and acknowledging that the findings may not be generalizable to the global population.

First, the manuscript addresses the prevalence of diabetes in the North-East Portuguese population, and I acknowledge the importance of this topic. However, in order to ensure its scientific rigor and potential impact, I believe that additional insights and clarifications are necessary.

In the article, the authors state: "current research confirmed a higher prevalence of T2D, IFG and diabetes risk than previous Portuguese epidemiological reports." While it is true that T2D prevalence is higher worldwide, I am particularly interested in understanding how the results differ from those reported in other studies and international guidelines. Therefore, I kindly request the authors to provide a specific discussion regarding the novelty and innovative contribution of this research in relation to existing literature.

Furthermore, the authors mention that future epidemiological reports should include the influence of sociodemographic factors, physical activity patterns, and health-risk behaviors to search for causal relationships in the prevalence of T2D, prediabetes, and diabetes risk. I would like to know if there are comprehensive guides or resources already available that cover this topic extensively.

Moreover, the authors claim that their results provide a more complete understanding of the prevalence of T2D, prediabetes, and diabetes risk in the North-East Portuguese population. To enhance the clarity of their contribution, I suggest that the authors provide a more specific discussion on how the findings relate to other papers and international guidelines on the subject. There are numerous guides available on diabetes, including country-specific resources such as the one from Portugal, as well as internationally recognized guides like the IDF Guide for Epidemiology Studies (https://diabetesatlas.org/idf-guide-for-epidemiology-studies/) and those published by renowned organizations such as the World Health Organization (https://www.who.int/health-topics/diabetes#tab=tab_1).

Regarding the presentation of results, I recommend some modifications. For instance, Figure 1 appears to be redundant as it only displays the prevalence of T2D for men and women within specific age groups, while all the information from this figure is already described in the text. The same redundancy is observed in Figure 2 (Prevalence of IFG for men and women within age groups) and Figure 3 (Prevalence of IFG in individuals without diabetes, by sex and age group). On the other hand, Figures 4 and 5 provide valuable information. However, it is unnecessary to repeat all the information from these figures in the text.

Thank you for your valuable contribution to the field.

Dear authors, 

Minor review of English language required.

Author Response

Dear Reviewer,

We are also thankful to the reviewers for reviewing the manuscript and suggesting valuable amendments for helping us to enhance its quality. We have revised the manuscript and providing a point to point response to all the comments. Also, we have highlighted all the amends on the manuscript.

Response to Reviewer 2 Comments

Point 1: However, I highly recommend that you incorporate additional statistical tests to improve both your confidence in the interpretation and the presentation of results. There are many epidemiological studies available in the literature about the Type 2 Diabetes topic, therefore, the additional analysis would highlight the novel contributions of your paper.

I also suggest addressing the limitations of the analysis and acknowledging that the findings may not be generalizable to the global population.

Response 1: Thanks for your comment. We understand the reviewer’s concerns so we carefully reply to each comment. First, the statistical tests applied are only intended to meet the study objective that was to analyze the prevalence of type 2 diabetes (T2D), impaired fasting glucose (IFG) and the diabetes risk in a middle-aged population from the Northeast of Portugal. For this reason, we have applied the following epidemological statistical: (1) categorical variables were expressed using counts (n) and proportions (%); (2) chi-squared test or fisher exact test were applied whenever appropriate (with Cramer’s V for effect size); (3) binary logistic regression (log-binary model), with an age and sex adjustments to calculate odds ratio (OR) and their 95% CI. These statistical tests are widely used in epidemiological studies of chronic non-communicable diseases (please, see Stewart, A. (2022). Basic statistics and epidemiology: A practical guide. CRC Press; reference 40). Also, the sample power is ensured (added in revised version). We have calculated “the sample size, with a margin of error of 0.05 and a 95% Confidence Interval (CI), the minimum sample size should be 383 individuals (G*Power, version 3.1.5.1 Institut für Experimentelle Psychologie, Düsseldorf, Germany)”.

Second, this research is an update to the epidemiological data in this specific population, because the latest data for this population was conducted in 2015 (PREVIAB study) and 2017 (INSEF study): “In Portugal, previous studies also detected a high prevalence of T2D, however the latest Portuguese epidemiological reports about diabetes were performed in 2010, 2015 and 2018 [23]”. “Based on global estimates of the prevalence of diabetes for 2030, this disease shows an exponential increase worldwide, particularly in developed countries such as Portugal [20]. Therefore, an update on the epidemiological situation of diabetes in the Portuguese population and it is essential to understand the specific epidemiological status of each particular region”.

Obviously further epidemological reports with national sample populations are crucial, however they also require an involvement of several health centres, something that has not been carried out in the Portuguese context for 6 years. In fact, the last national report reports to the Portuguese National Diabetes Observatory (OND) in 2017, which reported a prevalence of 13.6% for the Portuguese adult and older population (aged 20-79 years) (please see references 23), 41 and 44). Thus, it is important to update the data and understand the regional stauts (i.e., North-East Portuguese population) for care planning with exercise-based preventive programms: “In addition, more epidemiological reports with clinical data from other health centres across the country are needed to provide an update on the epidemiological situation of diabetes in Portugal, since the current research only updates data in the studied region.”.

Point 2: First, the manuscript addresses the prevalence of diabetes in the North-East Portuguese population, and I acknowledge the importance of this topic. However, in order to ensure its scientific rigor and potential impact, I believe that additional insights and clarifications are necessary.

Response 2: Thanks for your comment. We completely agree. We added the following text to the discussion (highlighted using the Microsoft Word® tracing tool in the manuscript):

“The Northeast region of Portugal is located inland. It is a region with low population density and little industrialization, when compared to the coast of the country. The total population has declined in recent years (from 2011 to 2021, less 759 residents for a total of 34,582), particularly in the 15-64 age group. However, the elderly population (>64 years) increased significantly (from 2011 to 2021, more 1,543 residents for a total of 9,748). (PORDATA https://www.pordata.pt/censos/resultados/emdestaquebraganca-446) (reference 41).

In the recent past, per capita income has also been decreasing in this region compared to the European average (78.6% in 2019, 76.4% in 2020 and 74% in 2021), with the north-eastern region of the country showing a lower per capita purchasing power compared to the national average. This reality favours the prevalence of a large number of families in a situation of economic and social vulnerability, aggravated by the con-finements motivated by the COVID-19 pandemic, with negative consequences for promoting people's health.” (PORDATA https://www.pordata.pt/censos/resultados/emdestaquebraganca-446) (reference 41).

Point 3: In the article, the authors state: "current research confirmed a higher prevalence of T2D, IFG and diabetes risk than previous Portuguese epidemiological reports." While it is true that T2D prevalence is higher worldwide, I am particularly interested in understanding how the results differ from those reported in other studies and international guidelines. Therefore, I kindly request the authors to provide a specific discussion regarding the novelty and innovative contribution of this research in relation to existing literature.

Response 3: Thanks for your comment. We understand the reviewers point view, however although it is a regional epidemological report. A total of 6570 individuals aged 18–102 years were analysed, between January 2019 and December 2020, among which 3865 women (57.4 ± 18.1 years) and 2705 men (60.0 ± 16.8 years). More specifically, it is a representative sample of this regional population extracted from a patienti’s clinical records of 18,890 individuals were analyzed, with 12,320 of them being excluded from the data analysis based on the selection criteria. Furthemore, we have compared the results to previous Portuguese studies, and found a trend towards increased prevalence of T2D and changes in insulin resistance (expressed by impaired fasting glucose). The main reason is the already reported need to update epidemiological data on the prevalence of diabetes in this population. As previous mentioned, this rearch is an update to the epidemiological data in this specific population, because the latest data for this population was conducted in 2015 (PREVIAB study) and 2017 (INSEF study). In the discussion, in the sub-chapters our data are compared to these studies "4.1. Prevalence of T2D", "4.2. Prevalence of IFG" and "4.3. Diabetes risk in individuals without diabetes" compare current results to the previous studies. Our data update previous data and emphasise the need to develop primary intervention measures for diabetes, developing multifactorial approaches to physical activity, exercise and altered lifestyle in primary health care: “Physical activity and exercise play a key role in primary and secondary prevention of diabetes, and therefore healthy lifestyle strategies should be implemented to increase their practice in cases of prediabetes [26]”. Notwithstanding, we have added some concerns about new epidemological reports using “… clinical data from others health care centres throughout to provide an update on the epidemiological situation of diabetes in Portugal, as the current research only updates the data in the region studied”.

Point 4: Furthermore, the authors mention that future epidemiological reports should include the influence of sociodemographic factors, physical activity patterns, and health-risk behaviors to search for causal relationships in the prevalence of T2D, prediabetes, and diabetes risk. I would like to know if there are comprehensive guides or resources already available that cover this topic extensively.

Response 4: We appreciate and understand this reviewer's comment. We deleated the folowing text from the manuscript: “Thus, further epidemiological reports should include the influence of sociodemographic factors, physical activity patterns and health-risk behaviours to search for causal relationships in the prevalence of T2D, prediabetes and diabetes risk [23–25].”.

We aded the folowing text with the limitations of the study: “The results of the present study should be interpreted with the following limitations: (i) this is an observational study of a specific region of Portugal, hence the results cannot be generalized to the rest of the Portuguese population or to other geographic regions; (ii) the lack of access to socio-demographic variables and objective data on the physical ac-tivity levels, limit a greater depth in the interpretation of the results.”.

We also added some information about the sociodemographic characteristics of the region at the beginning of the discussion, namely as mentioned in the answer to point 2 (above).

Point 5: Moreover, the authors claim that their results provide a more complete understanding of the prevalence of T2D, prediabetes, and diabetes risk in the North-East Portuguese population. To enhance the clarity of their contribution, I suggest that the authors provide a more specific discussion on how the findings relate to other papers and international guidelines on the subject. There are numerous guides available on diabetes, including country-specific resources such as the one from Portugal, as well as internationally recognized guides like the IDF Guide for Epidemiology Studies https://diabetesatlas.org/idf-guide-for-epidemiology-studies/ and those published by renowned organizations such as the World Health Organization (https://www.who.int/healthtopics/diabetes#tab=tab_1).

Response 5: Thank you very much for your comment. When we mention that this study enables a better understanding of the prevalence of T2D, we mean that this study serves to update the data, as the last study with data from this region was in 2018. Therefore, it allows us to investigate the trend in the evolution of T2D prevalence. Additionally, the magnitude of our sample is quite large, which enables us to better estimate the prevalence in this population. This knowledge can contribute as a baseline for local health policymakers to choose the most appropriate strategies for reversing the trend in the prevalence of T2D. This study complies with the IDF Guidelines, which emphasize the need for studies for the planning of care and preventive services. We believe that studies highlighting a more local and up-to-date reality can encourage greater proactivity among decision-makers in this matter.

In this context, we have added in the discussion (at the end of 4.1. Prevalence of T2D topic) another sociodemographic aspect that may justify, at least in part, the evolution of the disease in this particular territory: " On the other hand, the higher prevalence of T2D in the age group over 64 years old, associated with the fact that this population increased significantly in this region, despite the total population reduction, may also have contributed to the higher prevalence of T2D observed in the present study compared to those described above".

Point 6: Regarding the presentation of results, I recommend some modifications. For instance, Figure 1 appears to be redundant as it only displays the prevalence of T2D for men and women within specific age groups, while all the information from this figure is already described in the text. The same redundancy is observed in Figure 2 (Prevalence of IFG for men and women within age groups) and Figure 3 (Prevalence of IFG in individuals without diabetes, by sex and age group). On the other hand, Figures 4 and 5 provide valuable information. However, it is unnecessary to repeat all the information from these figures in the text.

Response 6: We agree with the reviewer. We deleted figures 1, 2 and 3 from the manuscript and changed the text with the presentation of the results relative to figures 4 and 5 (now 1 and 2), so as not to repeat the data contained in the figures. 

For more details please see the revised version manuscript.

Round 2

Reviewer 2 Report

Dear authors,

The comments were well addressed. Now, I think your paper is worth to be published.

Best regards

Dear authors,

You must confer the editing of the English language.